# Implementation of Australia's renewed cervical screening program: Preparedness of general practitioners and nurses

Farhana Sultana[1,2], Lara Roeske[3], Michael J. Malloy[1,2], Tracey L. McDermott[4], Marion Saville[3,4,5], Julia M. L. Brotherton[2,4] *

**1** Formerly VCS Population Health, VCS Foundation, East Melbourne, Victoria, Australia, **2** Centre for Epidemiology and Biostatistics, Melbourne School of Population and Global Health, University of Melbourne, Carlton, Victoria, Australia, **3** VCS Pathology, VCS Foundation, Carlton, Victoria, Australia, **4** VCS Population Health, VCS Foundation, East Melbourne, Victoria, Australia, **5** Department of Obstetrics and Gynaecology, University of Melbourne, Parkville, Victoria, Australia

* jbrother@vcs.org.au

**Data Availability Statement:** Source data files cannot be shared publicly per ethics approval

## Abstract

The National Cervical Screening Program (NCSP) in Australia underwent major changes on December 1st, 2017. The program changed from 2-yearly Pap testing for women aged 18–69 years to 5-yearly HPV testing for women aged 25–74 years including differential management pathways for oncogenic HPV 16/18 positive versus HPV non16/18 positive test results and the option of self-collection for under-screened women. We conducted a survey among cervical screening providers in primary care to assess their level of preparedness in undertaking cervical screening before (pre-renewal) and after (post-renewal) the new program was implemented. Surveys were conducted between 14th August and 30th November 2017 (pre-renewal) and 9th February and 26th October 2018 (post-renewal) among cervical screening providers who attended education sessions related to the new guidelines. Preparedness was assessed in three areas: 1) level of comfort implementing the new guidelines (7 questions), 2) level of confidence in their ability to convey information about the new guidelines (9 questions) and 3) level of agreement regarding access to resources to support implementation (11 questions). Proportions were calculated for each question response and pre- and post-renewal periods compared using generalised linear models. Open-ended questions related to anticipated barriers and ways to overcome barriers were also included in the questionnaires. Compared to the pre-renewal period, a higher proportion of practitioners in the post-renewal period were more comfortable offering routine screening to women ≥25 years (p = 0.005) and more confident explaining the rationale for not screening before 25 years (p = 0.015); confident explaining a positive HPV 16/18 (p = 0.04) and HPV non 16/18(p = 0.013) test result and were comfortable with not referring women with a positive HPV non 16/18 test result and low grade/negative cytology for colposcopy (p = 0.01). A higher proportion of Victorian practitioners in the post-renewal period sample were also comfortable (p = 0.04) and confident (p = 0.015) recommending self-collection to under-screened women and agreed that self-collection is a reliable test (p = 0.003). The most commonly reported suggestion was to provide information, education and communication materials to

conditions because data are potentially reidentifiable. Data can be requested through the Bellberry Ethics Committee (contact at bellberry@bellberry.com.au), who will require data to be handled appropriately, in consultation with the authors.

**Funding:** The authors received no specific funding for this work.

**Competing interests:** JMLB and MS are chief investigators of the NHMRC Centre for Research Excellence in Cervical Cancer Control (APP1135172) from which FS (formerly) & TLM receives salary support. JMLB, MS and LR are investigators of a trial of primary HPV screening in Australia (Compass) that has received a part funding contribution from Roche Molecular Systems, Ventana Inc. USA. This does not alter our adherence to PLOS ONE policies on sharing data and materials.

both patients and practitioners. Compared to the pre-renewal period, practitioners in the post-renewal period were better prepared to implement the renewed screening program. Healthcare providers require further support to implement the self-collection pathway.

## Introduction

The National Cervical Screening Program (NCSP) in Australia has resulted in a halving of cervical cancer incidence and mortality since it began in 1991 [1]. However, the emerging body of international evidence on the efficacy of primary HPV screening, as well as high coverage with HPV vaccination in Australia, prompted a structured critical examination of the NCSP in Australia known as the "Renewal" process. Based on the findings of this detailed evidence review and extensive modelling, in April 2014, it was recommended that Australia replace its 2-yearly cytology based screening program with 5-yearly primary HPV screening including partial genotyping for HPV 16/18 and direct referral of women test positive for HPV16/18 to colposcopy [2]. The renewed program is expected to further reduce incidence and mortality from cervical cancer by another 20–30% [3, 4]. Major changes to the program are detailed in **Table 1**. A new self-collection pathway for never- and under-screened women who decline clinician-sampling has been made available in the new program [2]. Effective implementation of this new self-collection pathway could make a critical difference for equity in the program. The renewed NCSP commenced on 1st December 2017, although without a fully functional national register or availability of a laboratory authorised to test self-collected samples. A single authorised laboratory based in Victoria (VCS Pathology) began testing specimens in January 2018.

General practice is the frontline for inviting and engaging women to participate in cervical screening. Data suggest that in 2015–2016 about 1.6 per 100 encounters in general practice were for a Pap smear [5] and the 5 yearly participation rate in the period 2012–2016 was

**Table 1. Key changes to the National Cervical Screening Program in Australia.**

| Key changes | Old NCSP 1991-Nov 2017 | New NCSP (implemented December 1, 2017) |
|---|---|---|
| Primary Screening test | Cervical cytology (Pap test) | Cervical Screening Test comprising HPV test with partial genotyping (identifies HPV 16 and 18 separate to other oncogenic HPV) Reflex liquid-based cytology for all HPV positive test results |
| Age range | 18/20*-69 years | 25–74 years |
| Screening interval | 2 yearly | 5 yearly |
| Registry support | Individual state and territory based registries | Single national register |
| Self-collection | Not available | Available to women at least 30 years of age who decline a practitioner-collected sample, and who are under-screened women (2 or more years overdue from their last screening test– 4 years for cytology and 7 years for HPV test) or who have never had a cervical screening test |
| Invitations and reminders | Recall and reminders | Invitations and reminders |
| Sample collection | Slide | Liquid based sample |

NCSP = National Cervical Screening Program

*women were eligible at age 18 or two years after first intercourse, whichever was later. The target age group for participation and routine reporting was 20–69 years.

estimated at 81.9% [1]. Despite this generally high level of engagement with the program, some women remain unscreened and are more likely to be diagnosed with cervical cancer [6]. Notably women resident in lower socioeconomic areas, Aboriginal and Torres Strait Islander women, and women from cultural and linguistically diverse backgrounds have lower screening participation due to a range of barriers, which can be structural, cultural or personal [7, 8]. In the new program, general practitioners (GPs) and other cervical screening providers continue to play a crucial role by explaining the changes to the cervical screening program to women, collecting liquid based cervical samples, facilitating self-collection (for eligible women), and managing patients according to the renewed follow-up and referral pathways.

Transition is a challenging period for what has been a very successful program [1]. Failure at the health services level to effectively adapt to the changes, and to successfully support and engage with the new program, could mean that projected improvements in equity of access and outcomes, and in further reducing the cervical cancer burden, may not be achieved.

In this study, we opportunistically collected information from general practitioners and nurses as a part of the Royal Australian College of General Practitioners (RACGP) accredited education activities in the lead up to Renewal (pre-renewal period) and thereafter (post-renewal period) in order to determine how prepared practitioners felt to undertake cervical screening under the new guidelines, and as quality improvement information for our training sessions. Alongside such educational workshops, other education for GPs about the renewed program was provided by government under contract by the National Prescribing Service and all general practitioners received a mailed out kit of information materials about the new program developed by government in the month immediately prior to Renewal (November 2017) (see website for these materials: http://www.cancerscreening.gov.au/internet/screening/publishing.nsf/Content/resources-menu?OpenDocument&CATEGORY=3Health+Professional+Resources-3&SUBMIT=Search). We aimed to identify any emerging issues that could compromise effective delivery of the Program, so that resources, training and systems changes could be developed and implemented appropriately and in a timely way.

## Materials and methods

### Study design

Two surveys were conducted, one before (pre-renewal) and one after the renewed program was implemented on December 1, 2017 (post-renewal). The pre-renewal survey was conducted between 14th August 2017 and 30th November 2017 and the post-renewal survey between 9th February 2018 and 26th October 2018.

### Study participants

General Practitioners (GPs) and nurse cervical screening providers in general practices, sexual health clinics and community clinics who had enrolled in and attended education sessions related to the new cervical screening program and who completed a pre-education survey as part of that education were eligible for this study.

### Study procedures

The surveys were incorporated as routine baseline pre-education activities, and a component of medical education organised by VCS Foundation in accordance with RACGP QI&CPD accreditation requirements, delivered in an identical manner at sessions for educational purposes in the pre and post renewal periods. It is usual practice to survey all participants about their baseline knowledge and reflections about their current practice, including what areas

they would like assistance with prior to the delivery of medical education and training. VCS Foundation is an accredited Royal Australian College of General Practitioners (RACGP) Education Activity Provider and VCS Liaison Physicians provided dedicated education sessions on the new cervical screening program to practitioners attending conferences (GP17 in Sydney, NSW, GP18 on the Gold Coast, Queensland, and RMA18 in Darwin, NT) or at their practice when enrolling their practice in the Compass trial or requesting an update of the new cervical screening program in Victoria or South Australia. The Compass trial is a randomised controlled trial comparing 2.5 yearly cytology based cervical screening with 5-yearly primary HPV screening in Australia. The trial is being conducted by the VCS Foundation in collaboration with the Cancer Council New South Wales. The main trial has been ongoing in Victoria since 2015 with more than 500 practices involved in recruitment and follow-up. Details of the trial have been published elsewhere [9].

General practitioners and nurses were asked to complete the survey as part of their preparation for attending the education session, either beforehand or at the start of the session, in paper or electronic format (via Survey Monkey®). All questionnaires were completed anonymously and could not be linked back to the participant. An opening statement in the questionnaire explained the purpose, time required to complete the questionnaire (approximately 15–20 minutes) and its anonymity. No incentives were provided to participants of this study. We piloted the survey questionnaire during its development with 10 practitioners by asking them to complete it and refined it based on feedback received. The main changes were to the questions seeking to understand what approximate proportion of patients were women under 25 and how many Pap tests practitioners usually did. We adjusted how these were asked (proportion rather than number and longer time period) based on difficulties practitioners had answering our initial questions.

## Survey instrument

We reviewed the framework of Michie et al [10] in considering the survey's development. The survey included statements and questions to assess preparedness (see **S1 File** **Survey**). Preparedness of the practitioners was assessed in three areas: 1) level of comfort with implementing the new recommendations (7 questions), 2) level of confidence about their ability to convey information about the new recommendations (9 questions) and 3) level of agreement regarding access to systems and resources to support the transition of the program (11 statements). Practitioners were asked to rate their level of comfort on a Likert scale of 1 (not at all comfortable) to 5 (extremely comfortable) and level of confidence on a scale of 1 (not at all confident) to 4 (very confident). Practitioners were also asked if they agreed, disagreed or didn't know whether they had access to systems and resources in place to support them during transition. The questionnaire also collected demographic information, assessed other routine preventive services offered by practitioners, and the practitioners' level of confidence, in the pre-renewal period, in discussing Pap testing with eligible women and women who are reluctant to screen. The survey also included open-ended questions for practitioners to comment on the anticipated key barriers to implementing the renewed cervical screening program in their practice and possible solutions. The study was approved by the Bellberry Human Research Ethics Committee, approval number 2018-08-715.

## Sample size

A total of 322 surveys (161 in each of the pre- and post-renewal periods) is sufficient to detect a difference of 16% between the pre- and post-renewal period in the various indicators (assessing level of comfort, confidence and access to resources) with 80% power and a 5% significance

level. This 16% difference is assuming 50% prevalence of the various indicators of preparedness in the pre-renewal period.

## Statistical analysis

All data were imported and analysed in STATA version 12.1 [11]. For categorical variables, frequencies and proportions were calculated whereas means and standard deviations were derived for normally distributed continuous variables. The pre- and post-renewal groups were compared for any difference in demographic characteristics using the chi-square test for categorical variables and t-test for continuous. Responses to questions related to level of comfort was dichotomised as 'comfortable' (including comfortable enough = 4 and extremely comfortable = 5) and 'not comfortable' (including not at all comfortable = 1, fairly uncomfortable = 2 and slightly uncomfortable = 3). Similarly, responses to level of confidence was dichotomised as 'confident' (confident enough = 3 and very confident = 4) and 'not confident' (not at all confident = 1 and not very confident = 2) and access to resources grouped as 'agreed' and 'not agreed/don't know'. Proportions were calculated for each question/statement (indicator) in the area of comfort, confidence and access to resources in the pre- and post-renewal period. To examine the change from the pre- to post-renewal period for each indicator (or outcome) in the three areas of preparedness (comfort, confidence and access), while adjusting for confounders, risk ratios were estimated using a generalised linear model with log link and a Poisson distribution with robust variance estimator [12]. A covariate (age, gender, place of practice, years and role in practice) was considered a confounder if it was associated with both the outcome and the exposure and not on the causal pathway between them. We also fitted interaction terms between each confounder and the exposure and reported stratum specific effects of the exposure on each outcome if there was evidence of an interaction. Number of cervical screening tests performed per month was not included in the model given that it is more likely to be a common effect of both the renewed program changes and preparedness of the practitioners. FS manually reviewed all qualitative responses and coded the data. Similar codes were grouped under themes and subthemes which were derived from the data and not identified beforehand. TLM independently reviewed the coding and the grouping of themes and subthemes and any discordant findings discussed and resolved by FS and TLM. Specific quotes were used to illustrate a survey finding where a need for better understanding was required.

## Results

A total of 395 surveys were returned. Of these, 71 (18%) were completed online. One respondent returned a blank questionnaire and 52 (13%) respondents were international medical graduates who were at different stages of their registration process in Australia and were therefore excluded from this analysis as non-representative of currently practicing Australian general practitioners and nurses. Of the remaining 342 surveys, 182 (53%) were from the post-renewal period. Compared with the pre-renewal period, the sample of practitioners in the post-renewal period were more likely to be general practitioners, from Victoria, <50 years of age and practicing for less than 10 years. There was a greater proportion of male practitioners in the post- than in the pre-renewal period sample (30% versus 17%, p = 0.005). While the average number of female patients seen per week was similar in the pre- and post-renewal period samples, the average number of cervical screening tests (Pap test in pre-renewal and cervical screening test in post-renewal period) performed per month was lower in the post- than in the pre-renewal sample period (p = 0.001, **Table 2**).

Lower proportions of practitioners self-reported routinely offering asymptomatic sexually active women aged <25 years cervical screening (p<0.001), other sexually transmitted

**Table 2. Characteristics of 342 Australian primary care practitioners recruited in the pre and post-renewal periods of the cervical screening program\*.**

| Characteristics | Pre-renewal (N = 160) | | Post-renewal (N = 182) | | P-value |
|---|---|---|---|---|---|
| | n | % | n | % | |
| **Mean age in years [min, max]** | **156** | 49 [23, 73] | **179** | 47 [26, 78] | |
| **Age categories** | **156** | | **179** | | |
| <50 years | 73 | 46.7 | 104 | 58.1 | 0.03 |
| 50+ years | 83 | 53.2 | 75 | 41.9 | |
| **Place of practice** | **153** | | **164** | | |
| Victoria | 51 | 33.3 | 95 | 57.9 | <0.001 |
| Other States^Ψ | 102 | 66.7 | 69 | 42.1 | |
| **Gender** | **158** | | **179** | | |
| Female | 131 | 82.9 | 125 | 69.8 | 0.005 |
| Male | 27 | 17.0 | 54 | 30.1 | |
| **Role in practice** | **160** | | **180** | | |
| GPs | 128 | 80.0 | 158 | 87.7 | 0.05 |
| Other (specify)^¥ | 32 | 20.0 | 22 | 12.2 | |
| **Years in practice categories** | **149** | | **181** | | |
| <10 years | 47 | 31.5 | 78 | 43.1 | 0.03 |
| 10+ years | 102 | 68.5 | 103 | 56.9 | |
| **Number of female patients seen per week** | **152** | | **177** | | |
| <20 | 25 | 16.4 | 29 | 16.3 | 0.69 |
| 20-<40 | 37 | 24.3 | 40 | 22.6 | |
| 40-<60 | 45 | 29.6 | 48 | 27.1 | |
| 60-<80 | 28 | 18.4 | 29 | 16.3 | |
| 80-<100 | 8 | 5.2 | 12 | 6.7 | |
| 100+ | 9 | 5.9 | 19 | 10.7 | |
| **Average number of CSTs performed per month (SD)** | **152** | 14 (±11) | **176** | 9 (±9) | |
| **Number of CSTs performed per month^£** | **152** | | **176** | | |
| <10/ month | 57 | 37.5 | 98 | 55.7 | 0.001 |
| 10 or more/month | 95 | 62.5 | 78 | 44.3 | |
| **Confidence discussing Pap with eligible women^β** | **160** | | **173** | | |
| Extremely confident | 58 | 36.2 | 65 | 37.5 | 0.90 |
| Confident | 73 | 45.6 | 79 | 45.6 | |
| Somewhat confident | 24 | 15.0 | 22 | 12.7 | |
| Not at all confident | 5 | 3.1 | 7 | 4.0 | |
| **Confidence discussing Pap with reluctant women^β** | **159** | | **175** | | |
| Extremely confident | 43 | 27.0 | 52 | 29.7 | 0.57 |
| Confident | 84 | 52.8 | 91 | 52.0 | |
| Somewhat confident | 28 | 17.6 | 24 | 13.7 | |
| Not at all confident | 4 | 2.5 | 8 | 4.5 | |

N = Total sample size of cohort, n = sample size, SD = Standard deviation

\* Non responders removed from each item denominator

Ψ Others include South Australia (n = 75), New South Wales (n = 53), Western Australia (n = 12), Queensland (n = 23), Northern Territory (n = 4) and Tasmania (n = 4)

¥ Others include nurse (n = 38), medical student (n = 2), hospital GP trainee (n = 1), midwife (n = 1), RMO (n = 4), Observership (n = 4) and practice manager (n = 1), did not mention (n = 3)

£ CST = cervical screening test refers to 5-yearly HPV testing in the post-renewal period and 2-yearly Pap testing in the pre-renewal period

β Respondents in the post-renewal were asked about their level of confidence discussing Pap testing prior to 1st December 2017

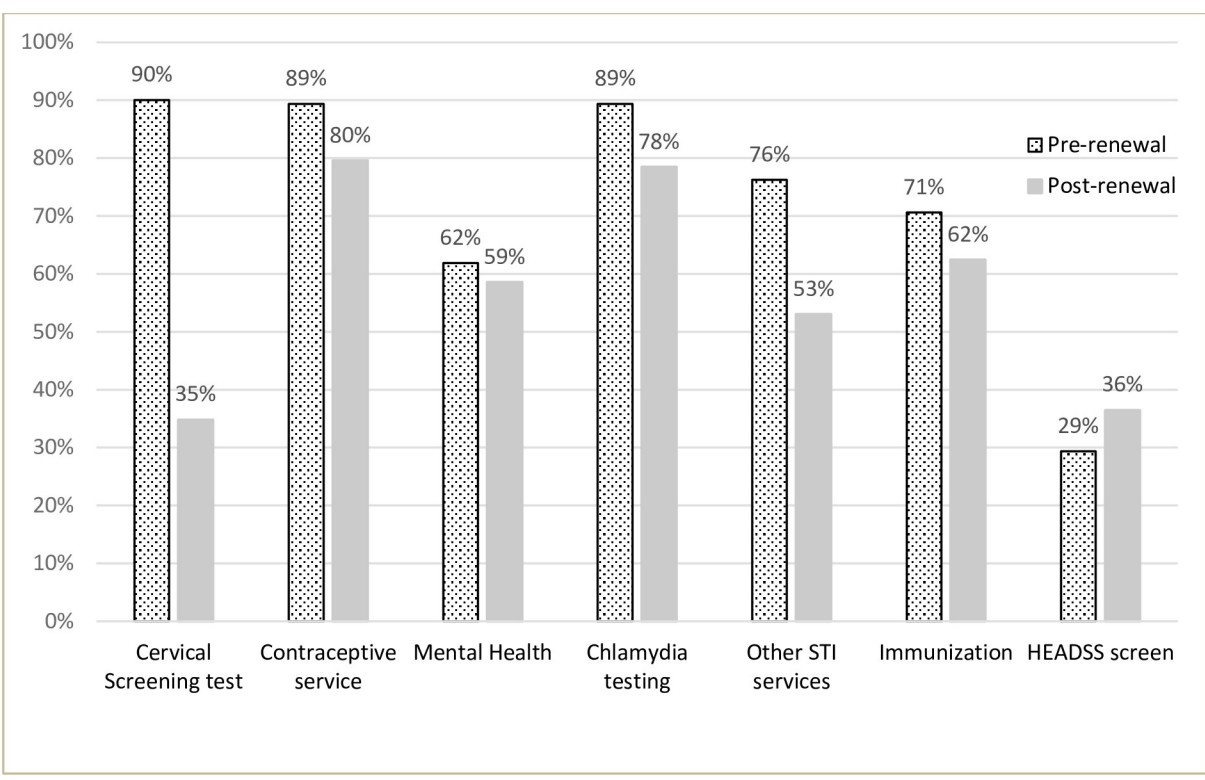

**Fig 1. Self-reported offering of preventive services to asymptomatic sexually active females <25 years in the pre- and post-renewal period samples by practitioners (n = 160 pre, n = 181 post).** HEADSS is a risk assessment tool to assess psycho-social health of adolescents and stands for Home, Education/employment, peer group Activities, Drugs, Sexuality, and Suicide/depression. Cervical screening tests are no longer routinely recommended to women <25 years in the post-renewal period with the exception of those in follow-up and some selected women (e.g. symptomatic).

infections (STI) testing (p<0.001), contraceptive services (p = 0.013) and chlamydia testing (p = 0.007) in the post- than in the pre-renewal period sample. There was no significant change in practitioner behaviour in the pre and post renewal periods in relation to provision of other services such as mental health (p = 0.53), immunization services (p = 0.11) or use of the HEADSS screening tool (p = 0.16) (**Fig 1**).

## Practitioner's comfort in implementing the renewed program

After adjustment for the differing characteristics of the two samples (confounders), practitioners in the post-renewal period, as compared to the pre-renewal period, were more likely to be comfortable offering routine screening only to women 25 years and over (p = 0.005) and not referring oncogenic HPV non 16/18positive women who have low grade/negative cytology for colposcopy (p = 0.01; **Table 3**).

With regards to practitioners' comfort with recommending self-collection to under-screened women, this varied by place of practice (p-value for interaction = 0.04). In Victoria, the proportion of practitioners who were comfortable recommending self-collection to under-screened women was found to be higher in the post- than in the pre-renewal period sample [70% versus 51%; RR: 1.38 (95% CI: 1.02 to 1.87); p = 0.04]. However, in the other states, this proportion was lower than in Victoria and slightly lower in the post- than in the pre-renewal period sample [37% versus 44% respectively; RR: 0.84 (0.57 to 1.23); p = 0.35]. The proportion of practitioners who were comfortable with women having to wait before being overdue to

**Table 3. Practitioners' comfort in implementing the renewed program in the pre- and post-renewal samples[*].**

| Key components | Questions/indicators | Pre-renewal | | Post-renewal | | Crude RR (95% CI) | P-value | Adjusted RR (95% CI) | P-value |
|---|---|---|---|---|---|---|---|---|---|
| | | n | % | n | % | | | | |
| **Age and interval** | 1. Only offering routine cervical screening to women 25 years and over | 108/ 156 | 69.2 | 151/ 175 | 86.3 | 1.25 (1.11 to 1.41) | **<0.001** | 1.20 (1.06 to 1.36) | **0.005**[α] |
| | 2. Screening HPV negative women every 5 years | 131/ 157 | 83.4 | 159/ 175 | 91.0 | 1.09 (1.00 to 1.19) | **0.04** | 1.05 (0.96 to 1.15) | 0.26[β] |
| **Sample collection** | 3. Collecting **only** a liquid based sample (not preparing a slide) | 139/ 157 | 88.5 | 158/ 174 | 91.0 | 1.03 (0.95 to 1.10) | 0.50 | 1.02 (0.95 to 1.10) | 0.57[£] |
| **Referral pathways** | 4. Referring all HPV16/18 positive women for colposcopy regardless of their cytology result | 117/ 153 | 76.5 | 148/ 175 | 84.6 | 1.11 (0.99 to 1.23) | 0.06 | 1.09 (0.99 to 1.21) | 0.08[£] |
| | 5. Not referring women with oncogenic HPV non 16/18 positive tests who have low grade/negative cytology for colposcopy | 63/ 152 | 41.4 | 108/ 173 | 62.4 | 1.51 (1.21 to 1.88) | **<0.001** | 1.35 (1.10 to 1.69) | **0.01**[Ψ] |
| **Self-collection** | 6. Recommending self-collection to an under-screened woman who refuses a practitioner-collected cervical sample | 73/ 156 | 46.8 | 101/ 176 | 57.4 | 1.23 (0.99 to 1.51) | 0.05 | 1.09 (0.87 to 1.37) | 0.44[€] |
| | 7. Having to wait to offer a repeat self-collection until the woman is overdue again (7 years since last screen) | 29/ 152 | 19.1 | 53/ 175 | 30.2 | 1.59 (1.07 to 2.36) | **0.023** | 1.30 (0.86 to 1.97) | 0.21[α] |

[*] Non responders removed from each item denominator

α Adjusted for place of practice

β Adjusted for place of practice, role and years in practice

£ Adjusted for role and years in practice

Ψ Adjusted for place of practice and role in practice

€ Adjusted for place of practice and an interaction between the renewal time period and place of practice (**p = 0.045**).

access self-collection was higher in the post- than in the pre-renewal sample (30% versus 19%); however, this was not significantly different by time period when adjusted for place of practice (p = 0.21).

## Practitioners' confidence to convey information about the renewed program

Comparing practitioners' confidence explaining the key changes and associated rationale, after adjustment for potential confounders, they were more likely to be confident post- than pre-renewal conveying the rationale for not screening before the age of 25 years (p = 0.015), explaining a positive HPV 16/18 result (p = 0.04), a positive HPV non 16/18 result (p = 0.013), and explaining to a woman who is not eligible for self-collection why that is the case (= 0.01). (**Table 4**). With regards to practitioners' confidence discussing self-collection with eligible women, this varied by place of practice (p-value for interaction = 0.028). In Victoria, a higher proportion of practitioners in the post-renewal group were confident discussing self-collection with an under-screened woman compared to the pre-renewal period group [70% versus 47% respectively; RR: 1.49 (1.08 to 2.08); p = 0.015]; however, this was not the case in other states [37% versus 43% respectively; RR: 0.85 (95% CI: 0.58 to 1.25); p = 0.41].

## Practitioners' access to resources

Practitioners were more likely to have access to educational materials and systems, as well as to information regarding renewal, in the post- rather than in the pre-renewal period for most categories after confounder adjustment. Notable exceptions were that there was no change in reporting by practitioners in their understanding of how to obtain information about their patients from the national cancer screening register (p = 0.79) or in their level of trust in the

**Table 4. Practitioners' confidence to convey information about the renewed program in pre- and post-renewal samples\*.**

| Key components | Questions/indicators | Pre-renewal | | Post-renewal | | Crude RR (95% CI) | P-value | Adjusted RR (95% CI) | P-value |
|---|---|---|---|---|---|---|---|---|---|
| | | N | % | n | % | | | | |
| **HPV based screening** | 1. Recommend HPV screening to a woman | 141/157 | 89.8 | 169/177 | 95.5 | 1.06 (1.00 to 1.13) | 0.05 | 1.06 (1.00 to 1.13) | 0.05[α] |
| | 2. Explain the association between HPV and cervical cancer to a woman | 147/156 | 94.2 | 168/177 | 94.9 | 1.01 (0.96 to 1.06) | 0.78 | 0.98 (0.93 to 1.03) | 0.47[β] |
| **Age and interval** | 3. Explain to a woman why more frequent cervical screening (i.e. every 2 years) is no longer recommended | 113/157 | 71.9 | 150/176 | 85.2 | 1.18 (1.05 to 1.33) | **0.004** | 1.12 (0.99 to 1.26) | 0.08[β] |
| | 4. Explain to a woman aged less than 25 years why she is not eligible for routine cervical screening | 96/155 | 61.9 | 141/175 | 80.6 | 1.30 (1.13 to 1.50) | **<0.001** | 1.21 (1.04 to 1.41) | **0.015[£]** |
| **Explaining HPV test results** | 5. Explain a negative HPV test result to a woman who will be asked to return in 5 years | 135/157 | 85.9 | 163/177 | 92.1 | 1.07 (0.99 to 1.16) | 0.07 | 1.03 (0.95 to 1.12) | 0.43[β] |
| | 6. Explain a HPV16/18 positive HPV test to a woman | 116/156 | 74.4 | 155/177 | 87.6 | 1.18 (1.06 to 1.31) | **0.003** | 1.12 (1.00 to 1.25) | **0.04[β]** |
| | 7. Explain a non 16/18 (other oncogenic) HPV positive test result to a woman | 95/156 | 60.9 | 139/177 | 78.5 | 1.29 (1.11 to 1.49) | **0.001** | 1.22 (1.04 to 1.44) | **0.013[Ψ]** |
| **Self-collection** | 8. Discuss the self-collection option with an eligible (under-screened) woman | 70/157 | 44.6 | 104/176 | 59.1 | 1.33 (1.07 to 1.64) | **0.010** | 1.14 (0.91 to 1.44) | 0.25[€] |
| | 9. Explain to a woman who is not eligible for self-collection why this is the case | 61/157 | 38.8 | 107/176 | 60.8 | 1.56 (1.24 to 1.97) | **<0.001** | 1.39 (1.08 to 1.78) | **0.01[£]** |

\* Non-responders removed from each item denominator

α Adjusted for role in practice

β Adjusted for place of practice and role in practice

£ Adjusted for place of practice

Ψ Adjusted for place of practice, role and years in practice

€ Adjusted for place of practice and an interaction between the renewal time period and place of practice (**p = 0.028**)

provider of the national cancer screening register with their patients' data (p = 0.63). The practitioners' agreement of their understanding of self-collection eligibility and reliability also varied by place of practice (**Table 5**). In Victoria, the proportion of practitioners who agreed that they understood which patients were eligible for self-collection pathway was 32% pre-renewal compared to 65% in the post-renewal period [RR: 2.03 (95% CI: 1.30 to 3.17); p = 0.002] whereas in other states there was no difference in the proportion between the pre- and post-renewal period samples [33% versus 33%; RR: 0.99 (95% CI: 0.63 to 1.53); p = 0.947]. Similarly, the proportion of practitioners who agreed that self-collection is a reliable test was greater in the post- than in the pre-renewal period sample in Victoria [52% versus 23%, respectively; RR: 2.33 (95% CI: 1.33 to 4.07); p = 0.003] but not in other states, where the proportion was considerably lower in the post- than in the pre-renewal period sample [12% versus 28%, respectively; RR: 0.43 (95% CI: 0.21 to 0.88; p = 0.021)].

## Perceived barriers

A total of 127 practitioners in the pre-renewal and another 125 in the post-renewal provided further information in the open ended questions related to barriers (S2 File). Many practitioners, in the pre-renewal period, perceived the 5-yearly screening interval to be a key barrier to the acceptability of the new program (n = 29). According to these practitioners, patients who were accustomed to the 2 yearly screening interval were anxious about the potential for cervical cancer to be more advanced at diagnosis in the 5 yearly screening program. A 67 year-old female GP from Victoria with more than 20 years in practice said *"Patients always ask why it is*

**Table 5. Practitioners' reporting about access to resources and systems to support the transition of the cervical screening program in the pre- and post-renewal samples\*.**

| Key components | Questions/indicators | Pre-renewal | | Post-renewal | | Crude RR (95% CI) | P-value | Adjusted RR (95% CI) | P-value |
|---|---|---|---|---|---|---|---|---|---|
| | | n | % | n | % | | | | |
| **Self-collection** | 1. I clearly understand which patients will be eligible for the self-collection pathway | 50/156 | 32 | 93/171 | 54.0 | 1.70 (1.30 to 2.22) | <0.001 | 1.40 (1.06 to 1.86) | **0.018**[α] |
| | 2. Self-collection is a reliable test [β] | 41/156 | 26.3 | 62/170 | 36.5 | 1.39 (1.00 to 1.93) | 0.05 | 1.15 (0.82 to 1.60) | 0.42[β] |
| **Educational materials** | 3. I know where to find the new guidelines (2016) for cervical screening | 106/158 | 67.1 | 145/172 | 84.3 | 1.26 (1.11 to 1.43) | <0.001 | 1.26 (1.11 to 1.43) | <0.001[μ] |
| | 4. I have access to educational materials to support my patients under the new program | 79/158 | 50.0 | 136/172 | 79.1 | 1.58 (1.33 to 1.88) | <0.001 | 1.45 (1.22 to 1.73) | <0.001[£] |
| | 5. Staff in my practice can easily access materials in the work place that support them in implementing the new program | 51/156 | 33.7 | 103/166 | 62.1 | 1.90 (1.47 to 2.45) | <0.001 | 1.65 (1.26 to 2.17) | <0.001[£] |
| | 6. I have patient information about the new screening program in my waiting area | 30/155 | 19.4 | 77/169 | 46.6 | 2.35 (1.64 to 3.38) | <0.001 | 2.39 (1.64 to 3.48) | <0.001[€] |
| **Systems and information** | 7. I understand in what way the reminder and recall systems in my practice will need to change under the new program | 78/155 | 50.3 | 131/170 | 77.1 | 1.53 (1.28 to 1.82) | <0.001 | 1.53 (1.28 to 1.82) | <0.001[μ] |
| | 8. I know how I will obtain information about my patients from the national cancer screening register | 47/155 | 30.3 | 67/169 | 39.6 | 1.31 (1.00 to 1.77) | 0.08 | 1.04 (0.76 to 1.43) | 0.79[£] |
| | 9. I know who to contact if I have questions about screening results and recommendations for my patients | 68/156 | 43.5 | 107/168 | 63.7 | 1.46 (1.18 to 1.81) | <0.001 | 1.30 (1.04 to 1.62) | **0.023**[∏] |
| | 10. I understand how the national cancer screening register will support the new program | 57/155 | 36.7 | 106/170 | 62.4 | 1.70 (1.34 to 2.15) | <0.001 | 1.63 (1.28 to 2.08) | <0.001[Ψ] |
| | 11. I trust the provider of the national cancer screening register with my patient's data[€] | 56/155 | 36.1 | 65/168 | 38.7 | 1.07 (0.81 to 1.42) | 0.63 | 1.07 (0.81 to 1.42) | 0.63[μ] |

\* Non responders removed from each item denominator

α Adjusted for place of practice and an interaction between the renewal time period and place of practice (**p = 0.024**).

β Adjusted for place of practice and an interaction between the renewal time period and place of practice (**p<0.001**)

μ No potential confounding by covariates

£ Adjusted for place of practice

€ Adjusted age categories and gender

∏ Adjusted for place of practice and years in practice

Ψ Adjusted gender

*5 years now. What if I get cancer by then*?*"* Practitioners also voiced concern that the increased screening interval could have potential adverse impacts on other health issues due to reduced screening frequency (e.g. STI checks, contraception, other preventive activities in GP e.g. blood pressure) as mentioned by a 52 year-old female GP from Victoria *"Concerned that we have less opportunity for screening diseases like chlamydia, skin checks, BP etc."*. Some other GPs felt there would be a loss of patient follow-up with the 5-year screening interval and that a very good recall system is essential. A 52 year-old female GP who is highly engaged in screening (30 or more Pap tests per month) mentioned *"Many people move house; 5 years recall might not be as effective"*. On the other hand, very few practitioners in the post-renewal period perceived the 5 years interval to be a barrier, and the few that did, quoted similar reasons such as missing out on other STI screening, fear of not screening frequently enough and the safety of the 5-year interval (n = 5). In contrast, one GP in the post-renewal period mentioned that *"some patients are welcoming the longer gap"*.

Concerns around starting screening at age 25 years were mostly raised by GPs in the pre-renewal period (n = 12), with very few GPs in the post-renewal period (n = 4) identifying this

as a barrier. One male GP, in the pre-renewal period, and in his early 30s mentioned, *"We have had a woman < 25 years with cervical cancer requiring hysterectomy"*. Another GP, in their early 60s queried, *"Is it appropriate for women who have had early intercourse?"* and what to do *". . ..in special situations like women <25 years who have previously had abnormal smears?"*. A GP, in the post-renewal period, with more than 30 years in practice mentioned *"Patients are not confident about the new program, especially with regards to the waiting time until age 25 years. This is because previous educational campaigns to them (and their mothers and grandmothers) said to start earlier"*.

The other most commonly reported potential barrier raised in both the pre- (n = 48) and the post-renewal (n = 40) samples was the lack of patients' understanding of the new program and its recommendations. The practitioners mentioned that there was a *"lack of understanding of the role of HPV"*, *"lack of patient knowledge about why the changes are happening"*, *"poor public advertising (in press)"*, *"incorrect information in the media"*, and *"confusion in public about the new program"*. One GP from New South Wales said *"Some women have heard little and think they don't need to come in at 2 years but at five"*. Furthermore, practitioners also mentioned that there were not enough materials provided to them that they could use to educate their patients. A 34 year-old nurse, in the pre-renewal period, from South Australia who performs an average of 20 Pap smears per month said *"Not feeling like I have enough support materials provided to me to implement the new program"*. Practitioners' lack of knowledge and understanding of the new guidelines also came up as an issue in the post-renewal period (n = 37). A GP with more than 30 years in practice in Victoria said *"Many women have been advised by medical professionals that they don't need a cervical screening test until 5 years after a previous normal Pap smear"*. The other areas of confusion were *"confusion regarding testing under 25 years with previous abnormal Pap smear"*, *"confusion between screening and doing certain tests in symptomatic patients"*, *"understanding the correct protocol for specific situations e.g. women with hysterectomy with possible high grade endocervical abnormalities"*, *"applying the rules to various groups of patients"*, and *"unclear on what results actually need closer follow up and action when deemed to be low risk but say GP needs to consider further action"*. Another GP in the post-renewal period mentioned *"a lot more women end up having colposcopy–this may lead to longer waits in the public system"*.

Some GPs also raised concern around implementing the new program without the screening register being ready and referred to it as a *"false start"*. The GPs understood that the reminder and recall systems in their practice would have to change but were not sure how to obtain information about their patient from the register. One GP from Queensland who completed the survey in the post-renewal period said *"There is no data from previous screening at my fingertips. Should improve with My Health Record"*[the Australian government's patient controlled electronic health record which is being implemented]. Some GPs mentioned that they required more information on how the data is going to be protected in the new register.

There were some concerns around eligibility in the new program, including the self-collection pathway. Practitioners were unsure about what to do with *"patients who are not sexually active"*, *"menopausal women"*, *"women who had early sexual intercourse"* and *"women with past abnormal Pap test"*. One GP mentioned, *"For women who currently attend regularly for cervical screening, I don't anticipate any barriers. The key barriers are for the women who are reluctant to participate in any cervical screening program"*. With regards to self-collection, practitioners' reported *"poor publicity campaign"* and confusion among women and practitioners. Some of the confusion about self-collection as mentioned by practitioners are *"who is eligible to self-test"* and concerns around the *"long waiting time for self-collection"*. One practitioner also mentioned that *"explaining that self-collection was only for a small group of patients will be challenging"* and women not understanding that a speculum examination is required if they

test positive, *"Patient's lack of understanding self-collection if positive will still mean they need a physical examination"*. Some practitioners in the pre (n = 8) and many practitioners in the post-renewal period (n = 28) acknowledged that change in general is difficult as it would mean "*changing what people are familiar with and habits*" and that it is a matter of time *"getting used to the new terminology and how the program works"*.

When asked about what the practitioner's source of information about the changes to the cervical screening program were, most mentioned government communications followed by medical media and other colleagues in both the pre- and the post-renewal period. Professional bodies, networks and journals also played an important role in conveying information about the new program (**Table 6**).

**Suggestions for overcoming barriers.** The most commonly reported suggestion in both the pre- and the post-renewal period was to provide information, education and communication materials to both patients and practitioners. The practitioners wanted *"waiting room handout information for their patients"*, to *"have access to good graphs and materials to show patients"* and *"take time to explain why the changes are being made, face to face if possible, and supported with online/printed information access"*. The practitioners also acknowledged that the information would have to be tailored for Aboriginal and Torres Strait Islander women in a way that suits local knowledge and community. Practitioners were of the opinion that attending GP conferences and various seminars/workshops would enhance knowledge and skills in the new era and that talks should be organised regarding the new program and that this should be done well in advance as pointed out by one GP in the pre-renewal period *"Resources should be available well ahead of time. The amount of lead time is ridiculous"*. Other than this, the practitioners also mentioned good communication skills as a way to overcome some of the barriers and used expressions such as *"communicating confidently"*, *"talk to each client sensitively"*, *"take time to educate"*, *"encourage people to change"*, *"explain to patients that the new system is safe"*. Updating practice-based systems and reminders were also mentioned by some GPs in the post-renewal period.

**Table 6. Source of information about the changes to the National Cervical Screening Program in Australia reported by practitioners in the pre- and post-renewal samples.**

| Source of information* | Pre-renewal (n = 160) | Post-renewal (n = 182) |
|---|---|---|
| Government communications including communications from NCSP, SACSR, SA Health | 56; 35% | 81; 45% |
| Medical media | 57; 36% | 82; 45% |
| Colleague | 52; 33% | 62; 34% |
| Professional bodies and networks (e.g. SHINE SA, NPS, RACGP, APNA, IRIS, SHA, VCS), attending talks, conferences, seminars, lectures | 40; 25% | 38; 21% |
| Journals | 27; 17% | 38; 21% |
| Website/online resources | 9; 6% | 4; 2% |
| Others (includes pathology labs and others not specified) | 11; 7% | 8; 4% |

*Multiple responses allowed

NCSP = National Cervical Screening Program; SACSR = South Australian Cervical Screening Register; SA Health = South Australia Health

Medical media refers to Australia's largest digital health media network; SHINE SA = Sexual Health Information Networking & Education South Australia; NPS = NPS MedicineWise; RACGP = Royal Australian College of General Practitioners; APNA = Australian Primary Health Care Nurses Association; IRIS = sexual health education provider; SHA = Sexual Health Australia; VCS = VCS Foundation

## Discussion

Our study found that a higher proportion of practitioners in the post-renewal period were comfortable with the key recommendations of the renewed program and were confident explaining the rationale for the later starting age (25 years), the longer screening interval (5 years instead of 2) and the differential management pathways following HPV 16/18 versus other oncogenic HPV detection. A higher proportion of practitioners in the post-renewal period also reported that they had access to education, information and communication materials around the program changes. An area of remaining need in relation to preparedness for implementing the renewed program was around practitioners' level of comfort and confidence with recommending self-collection, especially, practitioners outside of Victoria.

### HPV screening and referral pathways

In both pre- and post-renewal periods, around 90% or more practitioners reported they were confident recommending HPV screening to women and explaining the association between HPV and cervical cancer. This is perhaps not surprising given that the screening program comes some 10 years after the implementation of the National HPV Vaccination Program. GPs were highly involved in the vaccination program, especially between 2007 and 2009 when all young women up to the age of 26 were offered the vaccine through primary care. In 2008, only a year after the HPV vaccination program commenced, a national survey of GPs showed that 94% were comfortable discussing HPV with eligible clients and 97% were confident that they could deliver the vaccine [13]. In contrast, a survey of practitioners in Ireland found less certainty about HPV knowledge and in HPV related discussions with patients in relation to both screening and vaccination, which in Ireland was only a school based program [14]. The quadrivalent HPV vaccine, which protects against the two most oncogenic HPV types 16 and 18, was used in the Australian vaccination program, which may explain our practitioner's relative confidence in explaining HPV16/18 results and the need for referral to colposcopy. On the other hand, although an increase from baseline, about one-third of the practitioners in the post-renewal period sample were not comfortable with the management of other oncogenic HPV infection (non-16/18). Their greater uncertainty regarding the appropriate management of other oncogenic HPV (non-16/18) infections may reflect a lack of familiarity with these types and their related risks as well as their relatively more complex pathway. The 2016 cervical screening guidelines states that a woman positive for oncogenic HPV (non-16/18) with a cytology report of negative or low grade (indicative of an acute productive HPV infection) should have a follow-up HPV test in 12 months with referral for colposcopy only if the follow-up HPV test is positive for oncogenic HPV (any type), demonstrating that the infection is persistent [15].

### Renewed screening age and interval

In our study, the majority of the practitioners in the post-renewal period were comfortable offering routine screening starting at age 25 years (86%) and screening women every 5 years (91%). These findings were similar to a previous Australian 2015 study, where 84% of GPs and 75% of nurse practitioners were willing to start screening at 25 years and screen women every 5 years if the national guidelines recommended it [15]. In that study, 90% of the practitioners mentioned that they would feel more comfortable starting screening at 25 years if a woman had received a full HPV vaccine course prior to onset of sexual activity [16]. A small proportion in both studies were not comfortable with a later starting age and extended screening interval. The open-ended responses in our study indicated that practitioner concerns included cervical cancer being missed in women less than 25 years and patient perceptions that cervical

cancers could become more advanced in the 5-year screening interval. Similar concerns were found in a US study, where practitioner reported barriers to extending screening intervals included patient concerns about missing cancer and concerns about liability [17]. Another main concern raised by practitioners with the 5-year screening interval was that women may not attend for other health checks (e.g. STI screening) in the absence of more frequent cervical screening. This was also raised as a concern in both the earlier Australian and the US study [16, 17]. Although a decrease from pre-renewal period, the majority of the practitioners, in the post-renewal period, in our study, reported offering chlamydia testing (78%) and a reasonable proportion reported offering other STI services (53%) to asymptomatic sexually active women <25 years. A 2007–8 Australian study reported that, whilst 85% of females aged 16–29 years attended at least one GP consultation per year, only 12% were tested for chlamydia and 50% attended cervical screening in the 20–29 years age group [18]. The higher chlamydia testing rates in our study suggest that we may have sampled a highly motivated, engaged group of practitioners, as would be suggested by their recruitment at education sessions and possibly a degree of over reporting. It is also unclear why 35% of the practitioners in the post-renewal period in our study reported offering cervical screening to asymptomatic women less than 25 years.

## Self-collection pathway

The proportion of practitioners who were comfortable and confident recommending self-collection to under-screened women and who agreed that self-collection is a reliable test increased from the pre- to post-renewal period; however, in our study, this increase was only seen amongst Victorian practitioners. Self-collection has been extensively evaluated in a number of different settings and has been found to be as accurate as practitioner-collected samples for detection of CIN2+ lesion when a PCR based test is used [19]. This is the first time that self-collection has been made available to under-screened women in the National Cervical Screening Program in Australia. Because no HPV assay currently includes the use of self-collected samples on its product indication, there was lack of clarity around how laboratory testing would be regulated as a part of the program, leading to the delay in provision of self-collection services in the renewed program, compounded by a lack of information about it in program materials which continued even once the test was available. VCS Pathology (in Victoria), was the first laboratory in Australia and the only laboratory during the study period accredited to test self-collected samples because each laboratory is required to undertake an off label in house validation study [20]. Although VCS Pathology made the testing available nationally free of charge to eligible women and practitioners under the program, through either direct mail of specimens from practitioners or via their local laboratory forwarding the specimen, the program did not promote this availability). This could possibly explain the improved preparedness of Victorian practitioners in providing self-collection services in our study, together with the reluctance of other pathology services to promote testing through another service provider. To date one further pathology laboratory has been approved to process self-collected samples.

## Strengths and limitations

The primary limitation of our study is that it was opportunistic and recruited practitioners registered to attend accredited education sessions related to Renewal or visited at their practice for education, who may be more likely to be engaged in cervical screening and therefore not representative of all practitioners in Australia. This would imply that the preparedness of these practitioners was likely greater than that of most other practitioners. The nurses were a small

proportion of the sample, so experiences were mostly limited to GPs. In 2015–2016, the mean age of the GPs in Australia was 52 years, 45% were female and around 57% had 20 or more years of experience in general practice [5]. The practitioners in our study were similar in age and experience; however, the majority were female. The strengths of this study include that participation was part of an education activity so near complete for attendees of structured sessions; the comprehensive survey instrument which captured many aspects of readiness for implementation; and its timeliness in collecting information from practitioners prior to and just after the program transitioned. Comparative data through ongoing surveys of practitioners will provide further information around practitioner preparedness as the program matures.

In summary, practitioners in our study felt prepared to deliver the program more confidently and with the necessary resources to implement the cervical screening program over time, with the exception of self-collection. Ideally practitioners may be better equipped for program transition if education and training of both practitioners and the community starts earlier than the month before transition. Registry services, to support safe and effective transition, need to be available at the point of implementation and not afterwards. We recommend, to other countries transitioning to HPV based screening, that regulatory issues relating to HPV testing are identified and dealt with prior to implementation. For programs including self-collection, practitioners require education and training about eligibility, availability, reliability and relevance of self-collection parallel to mainstream practitioner-collected cervical screening, with strong and clear messaging around self-collection so that practitioners feel comfortable and confident ordering self-collection for eligible women.

## Supporting information

**S1 File. Survey.**
(PDF)

**S2 File. Major themes and subthemes from open-ended responses.**
(DOCX)

## Acknowledgments

The authors acknowledge VCS Liaison Physicians Wendy Pakes, Stella Heley and Alexis Butler, and Maryanne Marin for their support with data collection.

## Author Contributions

**Conceptualization:** Farhana Sultana, Lara Roeske, Marion Saville, Julia M. L. Brotherton.

**Data curation:** Lara Roeske, Tracey L. McDermott.

**Formal analysis:** Farhana Sultana, Michael J. Malloy, Tracey L. McDermott.

**Methodology:** Julia M. L. Brotherton.

**Project administration:** Tracey L. McDermott.

**Supervision:** Julia M. L. Brotherton.

**Validation:** Julia M. L. Brotherton.

**Writing – original draft:** Farhana Sultana.

**Writing – review & editing:** Farhana Sultana, Lara Roeske, Tracey L. McDermott, Marion Saville, Julia M. L. Brotherton.

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
