## [Decision Letter · Decision Letter 0]

31 Oct 2019

PONE-D-19-24843

Implementation of Australia’s Renewed Cervical Screening Program: Preparedness of General Practitioners and Nurses

PLOS ONE

Dear Associate Professor Brotherton,

Thank you for submitting your manuscript to PLOS ONE. After careful consideration, we feel that it has merit but does not fully meet PLOS ONE’s publication criteria as it currently stands. Therefore, we invite you to submit a revised version of the manuscript that addresses the points raised during the review process.

Please submit a point-by-point response that addresses each of the comments raised in the reviews. Please pay special attention to the reviewer comments requesting clarification on your study methods.

We would appreciate receiving your revised manuscript by Dec 15 2019 11:59PM. To enhance the reproducibility of your results, we recommend that if applicable you deposit your laboratory protocols in protocols.io, where a protocol can be assigned its own identifier (DOI) such that it can be cited independently in the future. For instructions see: http://journals.plos.org/plosone/s/submission-guidelines#loc-laboratory-protocols

We look forward to receiving your revised manuscript.

Kind regards,

Erin Bowles

Academic Editor

PLOS ONE

Journal Requirements:

3. In your Methods section, please provide additional information about the participant recruitment method. Please ensure you have provided sufficient details to replicate the analyses such as: a) a description of how participants were recruited, and b) specific descriptions of where participants were recruited and where the research took place.

JMLB and MS are chief investigators of the NHMRC Centre for Research Excellence in Cervical Cancer Control (APP1135172) from which FS (formerly) & TM receive salary support. JB, MS and LR are investigators of a trial of primary HPV screening in Australia (Compass) that has received a part funding contribution from Roche Molecular Systems, Ventana Inc. USA.

We note that you received funding from a commercial source: Roche Molecular Systems, Ventana Inc.

Reviewers' comments:

Reviewer's Responses to Questions

**Comments to the Author**

1. Is the manuscript technically sound, and do the data support the conclusions?

Reviewer #1: Yes

Reviewer #2: Yes

2. Has the statistical analysis been performed appropriately and rigorously? 

Reviewer #1: Yes

Reviewer #2: Yes

3. Have the authors made all data underlying the findings in their manuscript fully available?

Reviewer #1: Yes

Reviewer #2: Yes

4. Is the manuscript presented in an intelligible fashion and written in standard English?

Reviewer #1: Yes

Reviewer #2: Yes

5. Review Comments to the Author

Reviewer #1: This is an important paper that provides a great breadth of detailed and subtle information that contributes valuable information about key self-collect HPV deployment. There has clearly been great effort made to collect and analyze the data, with overall strong methods employed, but there is key missing methodological information hindering fully interpreting the findings.

1. Please briefly provide an overview or more detail provider education and outreach in support of the NCSP, especially pre- and post-"renewal". E.g., how were materials developed, at what frequency were they deployed, were they uniformly deployed and accessed (rural vs. urban)? All deployments have differences that do not translate to other settings and populations, but in the quickly evolving and important area of self-collect sampling for cervical cancer screening, greater detail in education and outreach would provide greater possibility for potential cross-setting sharing of generalizable tools.

2. Since you emphasize delays and barriers due to laboratory validation delays, it might be useful to include comments/analysis accounting for regions with unanticipated delays in laboratory validation.

3. Please provide more recruitment/sampling details at pre AND post surveys to allow better understanding of why this is an acceptable/representative sample. For example, you mention differences in pre-post participant characteristics, but do not comment in detail on why they are adequate for pre/post-comparison.

4. Page 6, line 117

What was "routine baseline evaluation"? Were these materials that may have primed subjects for your survey?

5. Page 7, lines 134-135

If applicable, please comment on any qualitative frameworks used and/or a bit more on what "refined it based on feedback received" entailed". A key missing component hindering many biomedical intervention behavioral survey comparisons and cross population analysis and implementation is lack of detail regarding instrument development and validation methods.

6. Page 9, lines 180-183

Please provide a bit more detail in open-ended response methods. For example, coding manual developed, independently coded by each reviewer, and discordant coding was adjudicated via... I appreciate your use of good methods for open-ended response coding, but think inserting the usual detail found in psychology studies would both encourage non-psych. researchers to learn of best practices and also provide clarity on the process.

7. Pages 19-23

When not using a validated instrument or established instrument framework, open-ended responses are an often critical area of information. There is tremendous valuable information in the responses, and more detail is needed to interpret open-ended responses, for example, total responders providing open-ended, reporting based on coded themes/subthemes, more compact reporting of frequencies, how it ties to other results, etc. As written and reported, it provides anecdotal findings. Tied with more detail in the coding methods, there appears to be potentially valuable information to justify an additional analysis, table, or supplemental table.

Reviewer #2: Dear authors,

This is a well written and clearly presented paper describing the qualitative and quantitative findings of a survey related to preparedness of health practitioners regarding the change in the Australian cervical screening program. Understanding practitioners’ perceptions, including their level of comfort and confidence in the change in screening recommendations, is important to ensure the success of the new program.

The design and the statistical analyses conducted were appropriate and well described. The interpretations and discussion of the findings were well supported by the findings. The conclusions highlight the usefulness of this work. I only have four very minor comments for the authors’ consideration.

Minor

• Introduction, Line 95: RACGP it is spelled out in the Methods, but should be spelled out first here.

• Method, Study procedures, Line 117+: It is not entirely clear if GPs and nurse cervical screening providers could only attend the education sessions if attending one of the included conferences or they were part of a new practice in the Compass trial, or whether other GP/nurse providers had other opportunity to attend the education sessions (and thus have the opportunity to be involved in the survey). How representative were the include GP/nurses likely to be of all GP/nurses at relevant health services? I know you are address this in the strengths and limitations, but I think the method could be clearer about exactly who was in the study and who had the opportunity to be in the study.

• Results: Given how different the pre- and post- sample are, I wonder if it would more appropriate to only report the adjusted estimates. Please consider presenting the unadjusted and adjusted estimates in the table, but only highlighting adjusted estimated in text.

• The structure of the report was a little confusing. I think the ‘results and discussion’ section is really a results section (with quant and qual findings), and then you could have the sections that are currently after the conclusion in a ‘discussion’ section, and then the conclusion last.

6. PLOS authors have the option to publish the peer review history of their article (what does this mean?). If published, this will include your full peer review and any attached files.

Reviewer #1: No

Reviewer #2: No

---

## [Author Response · Author response to Decision Letter 0]

13 Dec 2019

Response to reviewers

PONE-D-19-24843

Implementation of Australia’s Renewed Cervical Screening Program: Preparedness of General Practitioners and Nurses 

Journal Requirements:

1. When submitting your revision, we need you to address these additional requirements. Please ensure that your manuscript meets PLOS ONE's style requirements, including those for file naming. The PLOS ONE style templates can be found at http://www.journals.plos.org/plosone/s/file?id=wjVg/PLOSOne_formatting_sample_main_body.pdf

 and http://www.journals.plos.org/plosone/s/file?id=ba62/PLOSOne_formatting_sample_title_authors_affiliations.pdf

This had been amended as requested

This has been amended as requested

3. In your Methods section, please provide additional information about the participant recruitment method. Please ensure you have provided sufficient details to replicate the analyses such as: a) a description of how participants were recruited, and b) specific descriptions of where participants were recruited and where the research took place.

We feel that the methods are fairly explicit about these procedures already but have added further details in the Methods as requested. 

Although the questionnaires were completed anonymously and could not be linked to the practitioners, the sample size is small and unique to practitioners involved in the trial and those engaged in cervical screening and therefore potentially re-identifiable due to age, years of experience and location of practice (i.e state and postcode) and individual qualitative responses recorded in the survey database. Explicit permission from ethics (Bellberry Human Research Ethics Committee) for the study (2018-08-715) would have to be attained before sharing the dataset online. 

b) If there are no restrictions, please upload the minimal anonymized data set necessary to replicate your study findings as either Supporting Information files or to a stable, public repository and provide us with the relevant URLs, DOIs, or accession numbers. Please see http://www.bmj.com/content/340/bmj.c181.long for guidelines on how to de-identify and prepare clinical data for publication. For a list of acceptable repositories, please see http://journals.plos.org/plosone/s/data-availability#loc-recommended-repositories. We will update your Data Availability statement on your behalf to reflect the information you provide.

Please see response to point a, above.

JMLB and MS are chief investigators of the NHMRC Centre for Research Excellence in Cervical Cancer Control (APP1135172) from which FS (formerly) & TM receive salary support. JB, MS and LR are investigators of a trial of primary HPV screening in Australia (Compass) that has received a part funding contribution from Roche Molecular Systems, Ventana Inc. USA.

We note that you received funding from a commercial source: Roche Molecular Systems, Ventana Inc.

Reviewers' comments:

Reviewer's Responses to Questions

Comments to the Author

1. Is the manuscript technically sound, and do the data support the conclusions?

Reviewer #1: Yes

Reviewer #2: Yes________________________________________

2. Has the statistical analysis been performed appropriately and rigorously? 

Reviewer #1: Yes

Reviewer #2: Yes

3. Have the authors made all data underlying the findings in their manuscript fully available?

Reviewer #1: Yes

Reviewer #2: Yes

4. Is the manuscript presented in an intelligible fashion and written in standard English?

Reviewer #1: Yes

Reviewer #2: Yes

5. Review Comments to the Author

Reviewer #1: This is an important paper that provides a great breadth of detailed and subtle information that contributes valuable information about key self-collect HPV deployment. There has clearly been great effort made to collect and analyze the data, with overall strong methods employed, but there is key missing methodological information hindering fully interpreting the findings.

1. Please briefly provide an overview or more detail provider education and outreach in support of the NCSP, especially pre- and post-"renewal". E.g., how were materials developed, at what frequency were they deployed, were they uniformly deployed and accessed (rural vs. urban)? All deployments have differences that do not translate to other settings and populations, but in the quickly evolving and important area of self-collect sampling for cervical cancer screening, greater detail in education and outreach would provide greater possibility for potential cross-setting sharing of generalizable tools.

We have done this as requested in the introduction but please note there are currently no published references available to support this description/observations of the program roll out. 

2. Since you emphasize delays and barriers due to laboratory validation delays, it might be useful to include comments/analysis accounting for regions with unanticipated delays in laboratory validation.

We have added additional text in the discussion explaining this more fully and providing a newly published reference (Smith 2019). 

3. Please provide more recruitment/sampling details at pre AND post surveys to allow better understanding of why this is an acceptable/representative sample. For example, you mention differences in pre-post participant characteristics, but do not comment in detail on why they are adequate for pre/post-comparison.

We have added more detail as requested. The method of recruitment and the type of practitioners were identical between the two time periods so we are confident that with the statistical adjustments and analyses we have done that the pre-post comparisons are appropriate. 

4. Page 6, line 117

What was "routine baseline evaluation"? Were these materials that may have primed subjects for your survey?

No this just means refers to the standard practice for medical education training that participants are asked to think about their current state of knowledge coming into the session in the same way they are asked to reflect on what they have learnt and give feedback about the session afterwards. Clarification has been added to the text. 

5. Page 7, lines 134-135

If applicable, please comment on any qualitative frameworks used and/or a bit more on what "refined it based on feedback received" entailed". A key missing component hindering many biomedical intervention behavioural survey comparisons and cross population analysis and implementation is lack of detail regarding instrument development and validation methods.

We reviewed the Michie et al framework as background. We pretested the draft instrument with 10 GPs. The refinements we made were related to asking the practitioner to estimate the volume of and type of patients – we changed the time period from week to month and way of estimating from number to % because our draft measures were more difficult for GPs to estimate. We have added these details to the text.

6. Page 9, lines 180-183

Please provide a bit more detail in open-ended response methods. For example, coding manual developed, independently coded by each reviewer, and discordant coding was adjudicated via... I appreciate your use of good methods for open-ended response coding, but think inserting the usual detail found in psychology studies would both encourage non-psych. researchers to learn of best practices and also provide clarity on the process.

We have added more detail to statistical analysis as requested. 

7. Pages 19-23

When not using a validated instrument or established instrument framework, open-ended responses are an often critical area of information. There is tremendous valuable information in the responses, and more detail is needed to interpret open-ended responses, for example, total responders providing open-ended, reporting based on coded themes/subthemes, more compact reporting of frequencies, how it ties to other results, etc. As written and reported, it provides anecdotal findings. Tied with more detail in the coding methods, there appears to be potentially valuable information to justify an additional analysis, table, or supplemental table.

A supplemental table has been added with the themes and subthemes and reporting frequencies. We have also added some details to the results section. 

Reviewer #2: Dear authors,

This is a well written and clearly presented paper describing the qualitative and quantitative findings of a survey related to preparedness of health practitioners regarding the change in the Australian cervical screening program. Understanding practitioners’ perceptions, including their level of comfort and confidence in the change in screening recommendations, is important to ensure the success of the new program.

The design and the statistical analyses conducted were appropriate and well described. The interpretations and discussion of the findings were well supported by the findings. The conclusions highlight the usefulness of this work. I only have four very minor comments for the authors’ consideration.

Minor

• Introduction, Line 95: RACGP it is spelled out in the Methods, but should be spelled out first here.

Added

• Method, Study procedures, Line 117+: It is not entirely clear if GPs and nurse cervical screening providers could only attend the education sessions if attending one of the included conferences or they were part of a new practice in the Compass trial, or whether other GP/nurse providers had other opportunity to attend the education sessions (and thus have the opportunity to be involved in the survey). How representative were the include GP/nurses likely to be of all GP/nurses at relevant health services? I know you are address this in the strengths and limitations, but I think the method could be clearer about exactly who was in the study and who had the opportunity to be in the study.

We have clarified this. In short yes they had to sign up to attend a session, the survey was not open to “passers by”. 

• Results: Given how different the pre- and post- sample are, I wonder if it would more appropriate to only report the adjusted estimates. Please consider presenting the unadjusted and adjusted estimates in the table, but only highlighting adjusted estimated in text.

Thank you for this suggestion which we have adopted.

• The structure of the report was a little confusing. I think the ‘results and discussion’ section is really a results section (with quant and qual findings), and then you could have the sections that are currently after the conclusion in a ‘discussion’ section, and then the conclusion last.

We have changed the headings as suggested. 

6. PLOS authors have the option to publish the peer review history of their article (what does this mean?). If published, this will include your full peer review and any attached files.

Do you want your identity to be public for this peer review? For information about this choice, including consent withdrawal, please see our Privacy Policy.

Reviewer #1: No

Reviewer #2: No

---

## [Editor Report · Decision Letter 1]

7 Jan 2020

Implementation of Australia’s Renewed Cervical Screening Program: Preparedness of General Practitioners and Nurses

PONE-D-19-24843R1

Dear Dr. Brotherton,

We are pleased to inform you that your manuscript has been judged scientifically suitable for publication and will be formally accepted for publication once it complies with all outstanding technical requirements.

With kind regards,

Erin Bowles

Academic Editor

PLOS ONE
---

## [Editor Report · Acceptance letter]

9 Jan 2020

PONE-D-19-24843R1 

Implementation of Australia’s Renewed Cervical Screening Program: Preparedness of General Practitioners and Nurses 

Dear Dr. Brotherton:

I am pleased to inform you that your manuscript has been deemed suitable for publication in PLOS ONE. Congratulations! Your manuscript is now with our production department. 

With kind regards,

on behalf of

Dr. Erin Bowles 

Academic Editor

PLOS ONE